# Unsupervised Clustering and Ensemble Learning for Classifying Lip Articulation in Fingerspelling

**DOI:** 10.3390/s25123703

**Published:** 2025-06-13

**Authors:** Nurzada Amangeldy, Nazerke Gazizova, Marek Milosz, Bekbolat Kurmetbek, Aizhan Nazyrova, Akmaral Kassymova

**Affiliations:** 1Faculty of Information Technologies, L.N. Gumilyov Eurasian National University, Astana 010008, Kazakhstan; amangeldi_n_3@enu.kz (N.A.); b.kurmetbek@kmge.kz (B.K.); nazyrova_aye_1@enu.kz (A.N.); 2Department of Computer Science, Lublin University of Technology, 36B Nadbystrzycka Str., 20-618 Lublin, Poland; m.milosz@pollub.pl; 3Institute of Economics, Information Technologies and Professional Education, Zangir Khan West Kazakhstan Agrarion-Technical University, Uralsk 090000, Kazakhstan; kasimova.akmaral2020@gmail.com

**Keywords:** lip articulation, trajectory clustering, fingerspelling, unsupervised learning, multimodal recognition, visual speech

## Abstract

This paper presents a new methodology for analyzing lip articulation during fingerspelling aimed at extracting robust visual patterns that can overcome the inherent ambiguity and variability of lip shape. The proposed approach is based on unsupervised clustering of lip movement trajectories to identify consistent articulatory patterns across different time profiles. The methodology is not limited to using a single model. Still, it includes the exploration of varying cluster configurations and an assessment of their robustness, as well as a detailed analysis of the correspondence between individual alphabet letters and specific clusters. In contrast to direct classification based on raw visual features, this approach pre-tests clustered representations using a model-based assessment of their discriminative potential. This structured approach enhances the interpretability and robustness of the extracted features, highlighting the importance of lip dynamics as an auxiliary modality in multimodal sign language recognition. The obtained results demonstrate that trajectory clustering can serve as a practical method for generating features, providing more accurate and context-sensitive gesture interpretation.

## 1. Introduction

Limited access to information remains a significant barrier for millions of deaf people worldwide [1]. Recent advances in machine learning, particularly in the integration of gesture analysis and visual speech processing, present new opportunities to combine these modalities and improve the accuracy of signed language interpretation. Recognizing the inherently multimodal nature of communication between people with hearing and speech disabilities, this study highlights the need to consider not only hand movements but also the complex and often ambiguous patterns of lip articulation that are integral to signed words.

According to the World Federation of the Deaf, there are over 70 million deaf people globally [1]. In Kazakhstan, official records cite approximately 32,000 deaf individuals, though unofficial estimates suggest the number may be closer to 80,000. The United Nations Convention on the Rights of Persons with Disabilities [2] guarantees the legal right of deaf individuals to full access to information and communication across all domains, including healthcare, education, justice, and emergency services.

Lip articulation plays a crucial role in sign language recognition, complementing hand gestures and contributing significantly to meaning. Across various sign languages, lip movements function as an essential modality to enhance clarity, particularly when manual signs are visually similar. For instance, while core vocabulary across different sign languages may overlap by 60–80%, the phonetic influence of the spoken language on lip articulation introduces substantial variation in interpretation [3].

Some sign languages, such as Japanese Sign Language (JSL), rely heavily on lip articulation to convey precise meanings, while others like American Sign Language (ASL) place greater emphasis on facial expressions and body movement. Even among closely related languages such as British Sign Language (BSL) and Australian Sign Language (Auslan)—both derived from the BANZSL family—differences in the use of lip articulation are evident [4].

These linguistic and cultural dissimilarities in the role of lip articulation across sign languages represent the hardship of designing general-purpose recognition systems. Moreover, the inherently ambiguous and variable nature of audio-visual lip features poses a significant challenge for automatic interpretation [5,6,7,8,9,10,11,12,13]. Traditional methods that rely on direct feature extraction 14–21 often fail to capture the dynamic configuration of lip movements, especially in continuous or fingerprint sequences [14,15,16,17,18,19,20,21]. To address this case, this study proposes a structured approach based on unsupervised clustering of lip trajectories to identify consistent articulatory patterns that better support recognition tasks.

The authors propose a novel methodology that applies unsupervised clustering to lip trajectories, revealing stable articulatory patterns that may not be obvious from direct observation or raw feature extraction. The application of lip trajectory clustering, particularly when using methods that account for temporal dynamics, offers significant advantages over direct feature extraction from raw video data. First, such methods eliminate noise and compensate for interpersonal variations in articulation by grouping trajectories with similar dynamic characteristics despite differences in speed, amplitude, or performance style. Second, clustering reveals latent structures in the data—recurring articulatory patterns that often remain unobvious to visual analysis or classical neural network approaches. Third, cluster labels are highly interpretable: they can be visualized, tracked, and used as understandable input features for subsequent verification or classification.

## 2. Literature Review

### 2.1. Audio-Visual Speech Recognition (AVSR)

Advancements in speech and sign language recognition are strongly driven by developments in machine learning and computer vision. Within this domain, three key research areas stand out: Audio-Visual Speech Recognition (AVSR), lip reading, and Sign Language Recognition (SLR). AVSR specifically aims to enhance speech recognition performance by combining audio and visual inputs, leveraging the complementary nature of these modalities. Progress in AVSR has contributed to the development of multimodal architectures, innovative training techniques, and generalized models that address some of the fundamental limitations of unimodal systems. These advancements have paved the way for more robust and context-aware speech recognition technologies

### 2.2. Audio-Visual Speech Recognition

This section reviews key advancements in Audio-Visual Speech Recognition (AVSR), emphasizing the integration of multimodal inputs, the impact of noise on recognition accuracy, and strategies to enhance performance in challenging conditions. Recent studies have explored innovative approaches such as Momentum Contrast (MoCo) combined with Word2Vec for feature extraction under suboptimal acoustic environments [5,6,7,8,9,10,11,12,13].

Yang et al. investigated the effect of environmental noise on AVSR performance, proposing an MoCo + Word2Vec model [5]. Evaluated using the LRS2 and LRS3 datasets (Chung et al.), this approach improved the robustness of feature extraction, enhancing the system’s resilience to background noise [6,7,8]. These findings have direct implications for real-world AVSR applications, particularly in uncontrolled or noisy environments.

Further research by Ivanko et al. analyzed the evolution of AVSR architectures, comparing Recurrent Neural Networks (RNNs), Gated Memory Models (GMMs), and Transformer-based models [9]. A central challenge highlighted was the integration of heterogeneous input types—visual cues, motion, and audio—into synchronized, real-time systems. Key concerns included the need for diverse and speaker-independent datasets and the computational demands of managing high-dimensional multimodal inputs.

Generative Adversarial Networks (GANs) have also been applied in AVSR. Yin et al. demonstrated that GANs improved overall system efficiency and performance across LRS2 and LRS3 datasets [10]. GAN-based models facilitated multi-sensor input processing, reduced energy usage, and optimized feature extraction workflows. However, their study focused primarily on AVSR for AI-enabled Internet of Things (IoT) applications, leaving areas such as model adaptability and long-term stability open for further exploration.

The reliability of data inputs in multimodal AVSR systems has been addressed by Ryumin et al. [11]. Their research tested AVSR models under high-noise conditions and found that integrating visual and auditory inputs significantly reduced the Word Error Rate (WER) by 43% compared to audio-only systems. This result emphasizes the value of multimodal learning for improving AVSR accuracy and robustness.

In the context of assistive technologies, Bhaskar and Madathil explored AVSR systems tailored for individuals with hearing impairments [12]. Their findings indicate that optimizing AVSR to process speech with varying acoustic properties enhances accessibility and performance in diverse communicative settings, especially where audio signals are insufficient.

The application of AVSR in mobile and embedded systems has also been investigated. He et al. examined the feasibility of deploying GAN-powered AVSR on compact hardware platforms [13]. Their results showed that system architecture optimizations enabled efficient model operation on devices with limited computational resources, paving the way for broader implementation in mobile and edge environments.

Finally, the choice of datasets plays a crucial role in AVSR model training and generalization. LRS2 and LRS3 remain key benchmarks for evaluating AVSR performance. Across the reviewed studies, multimodal architectures such as Transformers, Conformers, and GANs continue to shape the field, offering scalable and adaptive solutions for integrating audio and visual signals. Research is increasingly turning toward AVSR systems that serve users with speech impairments and operate effectively across varied environments.

Future directions in AVSR research are expected to prioritize expanding multimodal datasets, improving model efficiency for deployment on mobile and embedded devices and developing adaptive systems capable of dynamic adjustment in real-world conditions.

### 2.3. Lip Reading

Recent advances in automatic lip reading have been largely driven by the development of sophisticated deep learning architectures, including 3D Convolutional Neural Networks (3D-CNNs), Long Short-Term Memory (LSTM) networks, and integrated multimodal pipelines. One emerging area of interest focuses on the extraction of temporal–spatial features through 3D spherical neural networks.

Wang et al. proposed a hybrid model that combines 3D-CNNs with Vision Transformers to enhance the extraction of temporal and spatial characteristics from facial images [14]. Their method, which applied machine learning techniques to identify key lip regions, significantly improved the precision of facial expression and content recognition. Evaluated on the LRW and LRW-1000 datasets, the model achieved accuracy rates of 88.5% and 57.5%, respectively.

A related study by Exarchos et al. introduced a hybrid framework integrating 3D-CNNs with LSTMs, designed to detect motion-related features and model the temporal dynamics of speech [15]. Tested on the MobLip dataset, this model achieved an accuracy of 87.5%, confirming its effectiveness under previously unseen conditions.

Similarly, He et al. developed the 3D-MouthNet-BLSTM-CTC model, which utilizes Bidirectional LSTM (BiLSTM) layers for temporal modeling and Connectionist Temporal Classification (CTC) for improving prediction accuracy [16]. Their experiments on the Oulu-VS2 dataset demonstrated a precision rate of 96.2%, underscoring the robustness of the approach.

In addition to neural network-based approaches, research has explored the integration of hand-crafted features. Tsourounis et al. introduced the SIFT-CNN architecture, which incorporates Scale-Invariant Feature Transform (SIFT) detectors with CNNs to enhance facial key point recognition, particularly in challenging visual environments [17].

The application of machine learning in biomedical contexts has also gained attention. Voutos et al. investigated the use of multimodal speech data to assist individuals with tracheomalacia, achieving a prediction accuracy of 95%, thereby demonstrating the potential of lip reading for users with limited verbal communication [18].

Generative models, particularly Generative Adversarial Networks (GANs), have also emerged as a powerful tool for enhancing lip-reading systems. Hameed et al. demonstrated that incorporating GANs into frequency-domain and GPU-based methods improved speech recognition performance, especially in low-resource environments [19]. GAN models tailored for individual users were further explored by Pham and Rahne, who developed a 3D-CNN-GRU model aimed at improving speech comprehension for non-native speakers. The model achieved accuracy rates of 87% for known languages and 63% for unfamiliar languages, highlighting the importance of language-specific customization [20].

The integration of multimodal data has also been shown to enhance speech recognition systems. Kumar et al. investigated audio-visual fusion using CNN-LSTM architectures for non-native speakers, reporting up to a 20% improvement in recognition accuracy over conventional machine learning techniques [21].

In summary, current research in lip reading emphasizes the application of 3D-CNNs, BiLSTM, Vision Transformers, and GANs for robust speech recognition. Key directions include real-time speech analysis, synthetic data generation, and the creation of customized language models. Future work is expected to focus on the development of generalized, scalable lip-reading systems for deployment in real-world environments.

### 2.4. Lip Reading in the Context of Sign Language Recognition

Although limited in number, recent studies combining lip reading and Sign Language Recognition (SLR) demonstrate meaningful progress, largely fueled by advancements in deep learning, multimodal systems, and machine learning techniques. However, most research continues to treat manual and non-manual components of sign language—such as hand gestures, facial expressions, lip articulation, and body movements—as separate modalities. This fragmented approach overlooks the fact that non-manual signals often play a vital role in conveying nuanced meanings and enhancing recognition accuracy.

One notable study addressing these challenges is by Thahseen et al., who developed an intelligent system to support communication among hearing-impaired students using Tamil Sign Language [22]. Their system included four key modules: text-to-sign translation, lip-reading-to-sign conversion, speech-to-sign processing, and gesture-to-text back-translation. Despite its high level of automation, the model lacked a true multimodal integration of manual and non-manual features, limiting its contextual robustness.

A more integrated approach was presented by Javaid and Rizvi, who proposed a multimodal architecture (MM-SLR) designed to simultaneously recognize both manual gestures and non-manual cues such as facial expressions and lip articulation [23,24]. Their model employed a modified YOLOv5 for spatial detection of hands and faces in video input, followed by C3D and LSTM networks for spatio-temporal feature analysis. This method enabled the fusion of gesture and lip-reading data, resulting in a significant improvement in interpretation accuracy.

Another promising direction is found in the work of Jebali et al., who developed a deep learning model based on a CNN-LSTM framework capable of processing both manual and non-manual elements [25]. Unlike earlier studies that relied on depth cameras or specialized sensors, their model utilized standard RGB video input, making it more accessible and applicable in real-world environments.

Collectively, these studies highlight the growing recognition of the importance of integrating lip reading into broader SLR systems. By acknowledging the interplay between manual and non-manual cues, especially lip articulation, these approaches offer more accurate and context-aware sign language interpretation. However, there remains substantial room for further development of unified, multimodal frameworks that fully exploit the synergy of these modalities.

Despite significant advancements in sign language and multimodal recognition systems, the question of which features are most critical for accurately recognizing letters of the dactylic (fingerspelling) alphabet remains insufficiently explored. In particular, there is a lack of empirical evidence regarding the role and significance of lip articulation in the identification of individual letters—especially in systems that do not rely on specialized hardware.

Building upon the gaps identified in the literature review, this study formulates the following research questions:

RQ1: How can video data, without the use of additional specialized equipment, be utilized to accurately extract visual features of lip articulation relevant to the recognition of dactylic alphabet letters?

RQ2: In what ways can clustering algorithms, followed by classification techniques, demonstrate that lip articulation features form coherent and distinguishable groups suitable for identifying individual letters in the dactylic alphabet?

RQ3: Do the resulting clusters of lip articulation features provide evidence that lip-reading cues are essential and informative for improving the recognition accuracy of the Kazakh dactylic alphabet?

These questions are designed to address current challenges in the field by exploring the integration of non-manual features—specifically, lip articulation—into recognition systems. The overarching aim is to develop more robust, accessible, and accurate multimodal approaches to fingerspelling recognition, particularly in low-resource or real-world conditions.

## 3. Materials and Methods

In this work, the classification of lip movements is not considered a separate task but an auxiliary module that clarifies, focusing on the identification and analysis of signs of lip articulation by showing letters pronounced with the fingers [26]. The method allows us to identify control patterns through visual observation. Among the Turkic languages, one of the richest and phonetically rich systems was chosen: the Kazakh dactyl alphabet, which has 42 letters [27]. For example, the Turkish alphabet has 29 letters [28] and the Azerbaijani alphabet has 32 letters [29]. The Kazakh alphabet features special phonemes that require precise depiction of the lips, making it particularly suitable for studying the dynamics of lip movements. True, the phonetic richness of the Kazakh dactyl alphabet and the visual complexity of articulation enable us to examine the stable patterns of lip movements in depth, and the solution can be applied to other sign languages.

Identifying visual features of lip movements during the pronunciation of dactyl signs is a difficult task, even for experienced observers. Due to the high variability of the articulation and the resolution of clear patterns, manual identification of signs is subjective. In this regard, this study used clustering methods (SoftDTW, DTW, etc.) to identify stable articulatory patterns based on the shape and dynamics of lip trajectories [30]. The cluster characteristics obtained served as the basis for the subsequent analysis, allowing an objective description of the internal structure of lip movements in dactyl speech.

A software architecture with step-by-step processing and processing was developed to create an analysis system. Manual identification of lip movements using visual means, due to their significant variability, primarily focused on clustering methods for feature extraction. The stages of processing and analysis of the section are as follows:Data collection and organization;Loading and grouping of .npy files;Extraction of trajectory features (distances between key points);Formation of temporal sequences;Extraction of features using clustering methods (SoftDTW, DTW, etc.);Visualization and evaluation of clusters (silhouette, trajectories).

As shown in the diagram, the main feature of the proposed approach is the use of clustering not for the purpose of identifying final features but as a means of identifying articulation patterns structured from the trajectories of lip movements. Visual observation of patterns enables us to characterize the internal dynamics of articulation resources and is especially important when working with phonetically dense alphabets.

### 3.1. Materials and Data Collection

To construct a high-quality dataset for Kazakh dactylic alphabet recognition, five professional sign language interpreters were recruited. Each interpreter performed 20 repetitions of each of the 42 letters, resulting in 100 video recordings per letter. In total, 4200 video samples were collected, forming a rich and diverse dataset suitable for training and evaluating machine learning models.

In general, the alphabet under study consists of 42 letters, among which there are signs of hardness and softness; these signs do not have a phonetic interpretation. Accordingly, when presented using fingerspelling, the shape of the lips does not change. When presenting the remaining 40 letters via fingerspelling, a change in lip shape is observed for each letter, corresponding to its sound in natural language.

To achieve the study’s goal, the authors identified 11 lip shapes by visually observing the changes in the lips when presenting these 40 letters. It should be noted that, since sign language interpreters were involved in the data collection process, there are differences between lip movements during the presentation of letters in sign language and those during natural speech articulation. This is because sign language interpreters tend to open their lips as widely as possible to make their signs more understandable to people with hearing impairments.

Letters were grouped according to each identified lip shape. The 11th group excluded dynamic letters, i.e., letters whose lip shape changes progressively as they are articulated. The remaining 10 groups of letters were organized as shown in Table 1—all steps prior to this grouping were performed manually based on the authors’ visual assessment.

### 3.2. Feature Extraction and Clustering

Extraction of lip movement features from the collected videos was carried out by automatically processing the time-dependent trajectories of facial points obtained from each frame of the video. The video was processed at 25 frames per second, and 30 frames were recorded for each letter from the beginning to the end of the lip movement. The list of sequences obtained was submitted to clustering using the TimeSeriesKMeans method with DTW (Dynamic Time Warping) and SoftDTW metrics. The number of clusters was tested starting from 10 and evaluated using silhouette coefficients.

Here, it should be noted that out of 4200 recorded videos, 200 videos corresponding to two letters (hardness sign and softness sign) were excluded because they did not affect the shape of the lips, and the remaining 4000 video files were submitted to clustering. During the analysis, various clustering configurations were considered, ranging from 10 to 4 clusters. The goal was to determine the number of clusters that best represent the structure of data on lip articulation while ensuring the stability, separability, and interpretability of the groups. Figure 1 presents silhouette graphs (left) and two-dimensional projections of feature spaces (right) for each configuration. These visualizations allow both quantitative and qualitative assessment of clustering results.

In the configuration of 10 clusters, high fragmentation is observed: several groups (for example, Clusters 4, 5, and 9) demonstrate good silhouette values and visual isolation, while Cluster 0 consistently shows negative values, indicating its internal heterogeneity. As the number of clusters decreases, there is a tendency for the average silhouette coefficient to increase. This suggests that larger groups have better internal consistency and minimize intersections between clusters. At the same time, visual projections confirm that the contours of clusters become clearer, and the distribution of points is dense and homogeneous.

The most balanced structure is achieved with a configuration of four clusters. Here, the most excellent coherence within groups, a high degree of distinguishability between clusters, and a minimum number of intersecting points are observed. This makes the model interpretable and stable during subsequent classification.

To quantitatively confirm the observed clustering patterns, a comparative table (Table 2) was compiled, reflecting the key characteristics of each configuration, ranging from 10 to 4 clusters. The table presents the values of the average silhouette coefficient and also identifies the best and problematic clusters based on both visual and numerical analysis. This allows for a more objective assessment of the dynamics of changes in the cluster structure as the number of clusters decreases.

As can be seen from the table, as the number of clusters decreases, a gradual increase in the average silhouette value is observed—from 0.38 for 10 clusters to 0.44 for 4 clusters. This indicates an increase in coherence within the groups. At the same time, the stability of some clusters (for example, 4, 5, and 6) is preserved, while Cluster 0 demonstrates a weak structure in all configurations.

The overall analysis indicates that the configuration with four clusters is the most balanced and suitable for subsequent classification. To determine which letters belong to each cluster, a comparative study was conducted using four clustering algorithms: Euclidean k-means, k-shape k-means, DTW k-means, and SoftDTW k-means. This allowed us to evaluate how different methods interpret the dynamics of lip movements, and the results showed that the SoftDTW method provides the most stable (Figure 2).

As shown in Figure 2, the clustering results reveal distinct groupings of lip movement trajectories based on articulatory similarity. Each cluster reflects distinctive temporal and spatial characteristics of lip movements associated with specific subsets of the finger alphabet. The following descriptions of the clusters provide a deeper insight into the composition and behavioral patterns observed in the four identified groups:Cluster 0: [‘A’, ‘AE’, ‘YA’] — includes gestures with short and rounded trajectories (see Figure 3).Cluster 1: [‘CH’, ‘E2’, ‘G2’, ‘H’, ‘NG’, ‘R’, ‘SH2’, ‘Y2’, ‘ZH’, ‘C’, ‘D’, ‘E’, ‘G’, ‘I’, ‘K’, ‘L’, ‘N’, ‘Q’, ‘S’, ‘SH’, ‘T’, ‘X’, ‘Y’, ‘Z’] — the largest group, characterized by medium trajectory length and the presence of several peaks.Cluster 2: [‘O’, ‘U’, ‘U2’, ‘U3’, ‘O2’, ‘YO’, ‘YU’] — contains mostly vowels that demonstrate smooth and extended movements (see Figure 4).Cluster 3: [‘B’, ‘P’, ‘P2’, ‘F’, ‘V’, ‘M’] — includes gestures with sharp, short movements and minor deviations (see Figure 5).

After all 40 letters were successfully distributed into four stable clusters, it was revealed that the largest cluster (Cluster 1) included 24 letters. This indicates significant internal diversity of gestures within this group. To further refine the structure and improve the accuracy of the recognition model, these 24 letters were resubmitted to the same experimental protocol but with a changed number of Cluster 2. That is, at the same time, trajectory clustering algorithms were re-applied to them but now to divide them not into four but into two subclusters.

This approach enabled us to examine the internal heterogeneity of the most saturated group in greater detail and, in the future, utilize these subgroups to develop a more accurate hybrid classification model. These two methods yielded similar results when grouping the 26 letters, leading to the following cluster distributions (see Figure 6 and Figure 7). The final analysis generated five clusters in total (Figure 3, Figure 4, Figure 5, Figure 6 and Figure 7), enabling a more refined classification of Kazakh alphabet letters Table A1 and enhancing the precision of the gesture recognition model.

### 3.3. Classification Algorithms

To verify the usefulness of cluster labels as informative features, various machine learning algorithms were applied. The goal of this stage was to determine whether automatically extracted lip movement patterns can serve as a reliable basis for recognizing the dactyl alphabet independently of manual gestures. Key metrics were used to evaluate the effectiveness of each algorithm: accuracy, robustness, and generalization ability on the test set.

For example, the Decision Tree classifier demonstrated 63% accuracy on the training dataset, indicating its fundamental ability to distinguish letters based solely on visual features of lip articulation (Figure 8).

The XGBoost classifier outperformed all other methods, achieving the highest accuracy on the training dataset (Figure 9). It also demonstrated strong performance in terms of stability and generalization on the test set. Thanks to its capacity to manage nonlinear relationships and complex feature interactions, XGBoost proved to be the more effective and appropriate algorithm, with an accuracy of 74%, for this classification task.

The Random Forest classifier achieved an improved accuracy of 75% (Figure 10), providing consistent and reliable performance across both the training and test datasets. Its ensemble nature allowed it to effectively capture data variability while reducing the risk of overfitting, contributing to its overall robustness.

The performance of all three classifiers confirmed the effectiveness of the clusters derived through unsupervised learning techniques, specifically DTW k-means and SoftDTW k-means. These clustering methods successfully grouped the letters of the Kazakh fingerspelling alphabet based on visual and articulatory features, thereby enhancing the accuracy of subsequent classification tasks. The Decision Tree classifier, Random Forest classifier, and XGBoost classifier all produced consistent results when applied to these clusters, indicating that the unsupervised clustering approach provided a reliable structure for the data. Among them, Random Forest achieved the highest performance, further demonstrating the model’s strong generalization capabilities on previously unseen data.

These findings emphasize the value of combining clustering with unsupervised learning methods as a strategy for improving classification accuracy. This approach plays a crucial role in developing gesture and lip movement recognition systems.

The clustering analysis of the Kazakh fingerspelling alphabet revealed the feasibility of effectively organizing gesture data into distinct groups based on both visual and articulatory features. Initially, 40 letters were grouped into 10 classes using a dataset enhanced with clear lip articulation. Silhouette analysis showed that Clusters 4, 5, 7, and 9 achieved the highest silhouette scores, reflecting strong separation and internal consistency. In contrast, Clusters 1, 2, 6, and 10 exhibited overlapping characteristics, signaling the need for further refinement. Through subsequent experiments, the optimal number of clusters was found to be four, resulting in the most stable and well-separated groupings with clearly defined boundaries and denser distributions.

To further improve model performance, 24 letters from one of the larger clusters were isolated and divided into two new classes based on shared lip articulation patterns. Clustering methods such as DTW k-means and SoftDTW k-means effectively distinguished these gestures by identifying subtle dynamic and temporal differences in their trajectories. This refinement ultimately produced five distinct clusters (Figure 5, Figure 6, Figure 7, Figure 8 and Figure 9), each containing letters that shared similar visual and motion-related characteristics.

For evaluation, the Decision Tree classifier achieved 62% accuracy on the training set, while the XGBoost classifier achieved 74%. Additionally, Random Forest demonstrated the highest accuracy and generalization ability at 75%. These outcomes validated the effectiveness of the unsupervised clustering process in structuring the data for classification.

Overall, the combination of clustering and classification techniques provided a structured and effective approach for organizing the Kazakh fingerspelling alphabet. This methodology not only improved recognition accuracy but also laid a solid foundation for future enhancements to the gesture and lip movement recognition system.

## 4. Results

This section outlines the results obtained at each stage of the experimental process, from data preprocessing to the final training of the multimodal recognition architecture.

In the initial phase, the collected video data was manually clustered through visual analysis of lip articulation features. This qualitative approach enabled the identification of 11 distinct classes, each grouping letters of the Kazakh fingerspelling alphabet with similar articulatory characteristics. Although the manual clustering process provided a valuable foundation for structuring the data, its inherent subjectivity highlighted the need for automated validation and refinement.

To address this, a second phase of automated clustering was performed using unsupervised machine learning algorithms. A range of clustering techniques—Euclidean k-means, KShape k-means, DTW k-means, and SoftDTW k-means—were applied to explore natural groupings within the data and evaluate the reliability of the initial manual partitioning.

Experiments were conducted using varying numbers of clusters (from 4 to 10) to determine the optimal level of separation. The results indicated Table 3 that increasing the number of clusters did not consistently improve cluster quality; in some cases, it led to overlapping classes and reduced separability. Among the tested algorithms, DTW k-means and SoftDTW k-means yielded the most coherent and meaningful partitions, with four clusters emerging as the optimal configuration in terms of clarity and interpretability.

These findings provided a robust foundation for the subsequent classification phase and confirmed that unsupervised clustering can effectively model articulatory similarities, thereby supporting the development of more accurate multimodal sign language recognition systems.

The optimal clustering solution was found by partitioning the data into four clusters, which offered a well-balanced combination of clear inter-cluster boundaries and high intra-cluster cohesion. However, subsequent analysis revealed that Cluster 1 contained 24 letters of the Kazakh dactylic alphabet, suggesting excessive internal heterogeneity.

To address this, Cluster 1 was further divided into two sub-clusters using the same experimental framework as the initial clustering—namely, the application of four clustering algorithms (Euclidean k-means, KShape k-means, DTW k-means, and SoftDTW k-means) across the subset of 24 letter groups, now with the number of clusters set to two. This resulted in a final configuration of five clusters.

This final five-cluster structure minimized classification errors and established an optimal data framework for multimodal training. It ensured a more accurate representation of articulatory features and enhanced the overall separability of the dataset. Following this clustering phase, classification was conducted using machine learning algorithms in Table 4 to verify the distinguishability of the data and confirm the validity of the identified groupings.

## 5. Discussion

Unlike supervised classification methods, which rely on pre-labeled datasets, clustering is particularly valuable in contexts where feature boundaries are ambiguous or difficult to define. This is especially true in the field of lip articulation analysis, where the visual characteristics of mouth movements are often subtle, highly variable, and not easily distinguishable by human annotators. In such cases, assigning accurate class labels for supervised learning becomes complex and prone to subjectivity.

To address this problem, this study presents an unsupervised clustering approach to lip movement trajectories, aiming to extract high-level, interpretable features. Instead of performing classification directly on raw video data or visual embeddings, as is typically performed in existing lip-reading [14,15,16,17,20,21] and visual speech recognition (VSR) studies [5,6,7,8,9,10,11,12,13], the proposed methodology uses time series clustering (using DTW and SoftDTW) to first structure the dynamic patterns of lip movement into semantically coherent groups.

This clustered representation allows us to

Reduce noise and variability between speakers;Detect hidden articulatory structures that direct observations may obscure;Moreover, create feature sets that are better suited for subsequent model verification or classification.

A comparative analysis reveals that, although many recent works employ deep learning models such as CNN [14,15,17,21], LSTM [9,16,23,24], or GAN [10,13,19] architectures directly on lip images or video frames, these approaches often lack interpretability. They are sensitive to variations in articulation styles. In contrast, the clustered pipeline enables the model to generalize data across different performers, emphasizing explainability—an increasingly important criterion in human-centric artificial intelligence systems.

Moreover, the results show that even when using relatively simple classifiers (such as Decision Tree, Random Forest, and XGBoost), the structured input obtained from clustering provides reliable recognition performance. This suggests that clustering not only improves interpretability but also contributes to classification accuracy when lip movements are used as the primary modality.

Thus, the proposed methodology addresses a critical methodological gap in VSR research by introducing uncontrolled clustering of lip trajectories as a separate and valuable pre-processing step. It builds on the author’s previous research in sign language recognition through hand gesture classification and extends it by integrating articulatory features as an additional channel [26,31,32]. This integrated perspective applies the basis for future developments in interpretable multimodal gesture recognition, illustrating that trajectory-based structuring can be a powerful alternative to purely end-to-end neural architectures.

## 6. Conclusions

This study addressed the problem of recognizing individual letters in the dactyl (finger) alphabet of sign language by studying the visual properties of lip movements. Authors propose a new approach that combines uncontrolled clustering of lip articulation features with subsequent classification using tree-based machine learning algorithms aimed at overcoming the limitations caused by the low variability and visual similarity of hand gestures representing letters.

In response to the initial research question, the study demonstrates that meaningful visual features of lip motions can be extracted from video data without the use of additional equipment. By employing clustering techniques such as DTW k-means and SoftDTW k-means, lip articulation patterns were successfully grouped into stable and informative clusters.

To assess the validity and discriminative power of these clusters, classification algorithms, including Decision Tree, Random Forest, and XGBoost, were applied. The results confirmed that the identified clusters were both separable and relevant for distinguishing specific letters of the Kazakh fingerspelling alphabet. These findings confirm the viability of lip articulation as a valuable source of features for enhancing recognition accuracy.

The results clearly indicate that lip articulation features contain discernible and informative cues suitable for recognizing dactylic letters. The use of clustering and classification based solely on lip movement introduces a promising direction for improving the precision and robustness of sign language recognition systems. This approach is particularly advantageous in cases where hand gestures alone fail to provide sufficient differentiation between letters, offering either an alternative or a complementary method to gesture-based recognition.

Further works will focus on expanding the dataset, refining the clustering techniques, and exploring additional classification models to further strengthen the role of lip-reading in automatic sign language recognition systems.

## Figures and Tables

**Figure 1 sensors-25-03703-f001:**
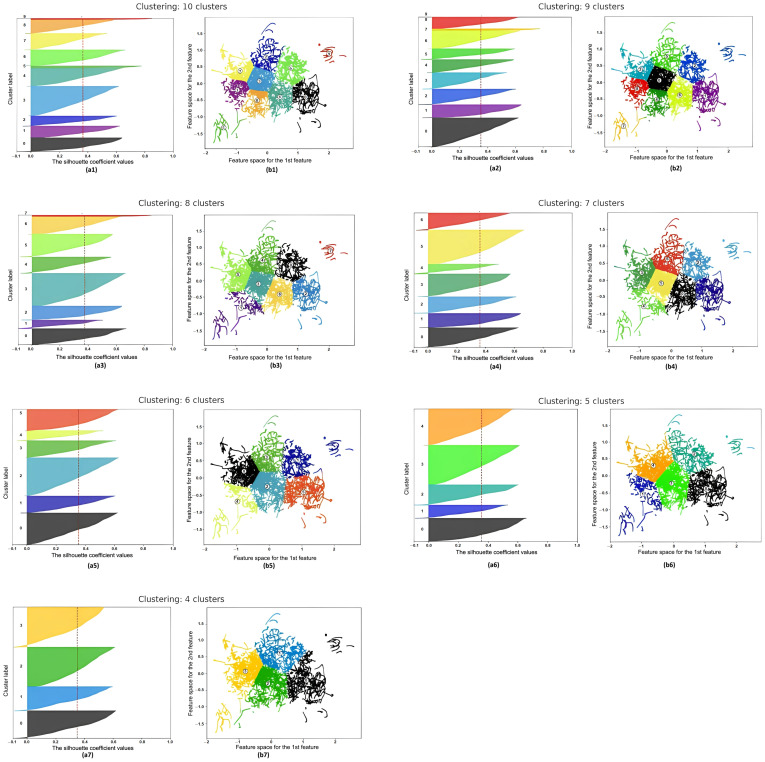
The silhouette plot for clusters (**a1**–**a7**) and the visualization of the clustered data (**b1**–**b7**).

**Figure 2 sensors-25-03703-f002:**
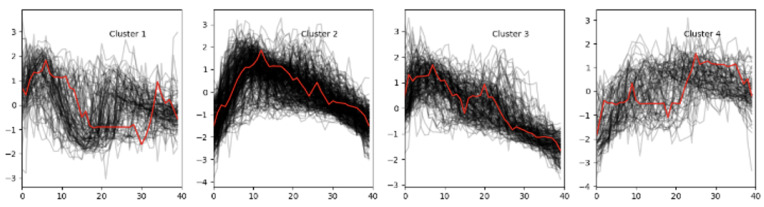
Cluster assignment of lip trajectories based on articulatory similarity.

**Figure 3 sensors-25-03703-f003:**
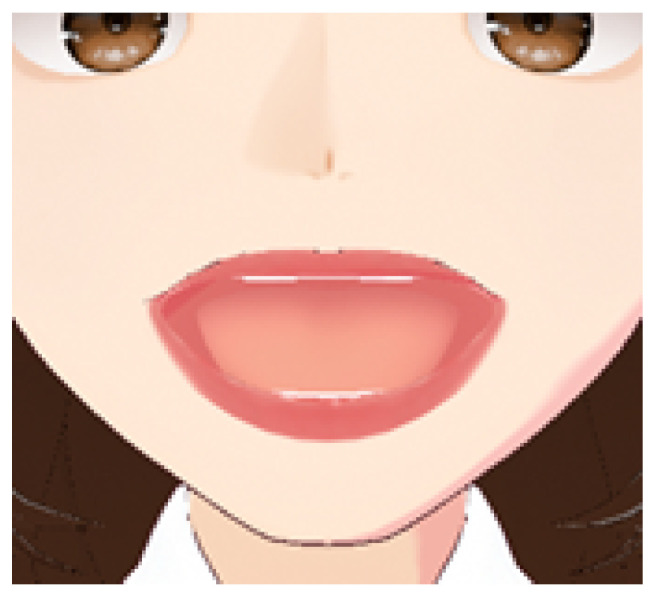
Cluster 0: Includes the letters ‘A’, ‘AE’, ‘YA’.

**Figure 4 sensors-25-03703-f004:**
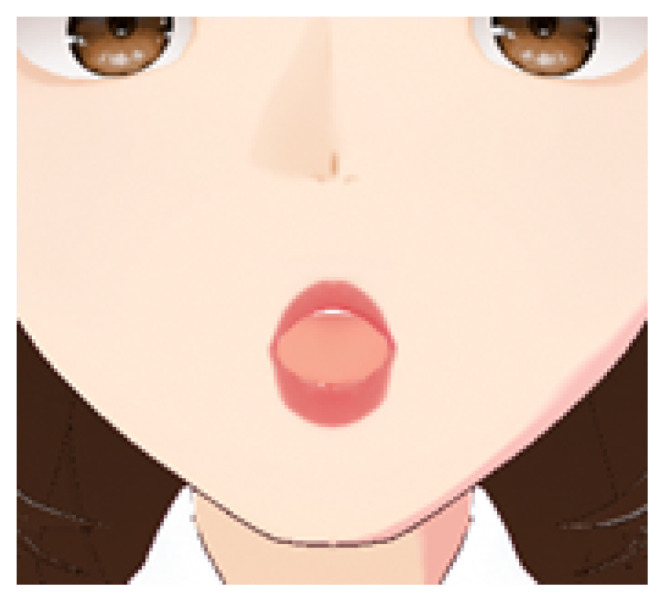
Cluster 2: Includes the letters ‘O’, ‘U’, ‘U2’, ‘U3’, ‘O2’, ‘YO’, ‘YU’.

**Figure 5 sensors-25-03703-f005:**
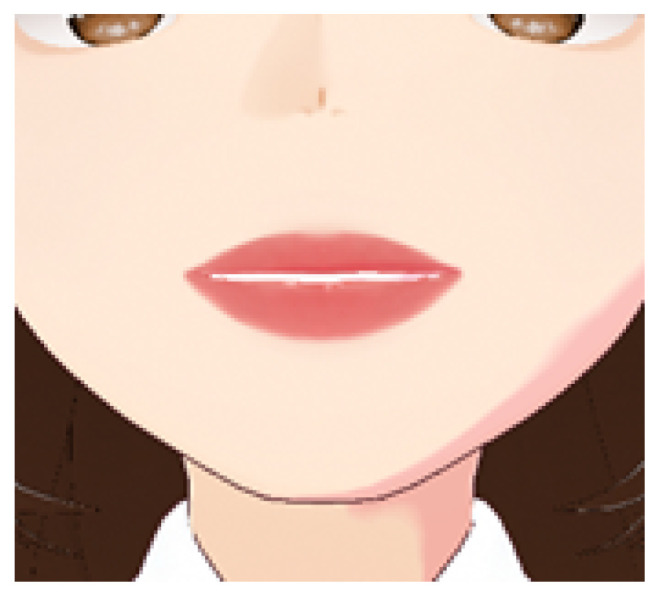
Cluster 1: Includes the letters ‘B’, ‘P’, ‘P2’, ‘F’, ‘V’, ‘M’.

**Figure 6 sensors-25-03703-f006:**
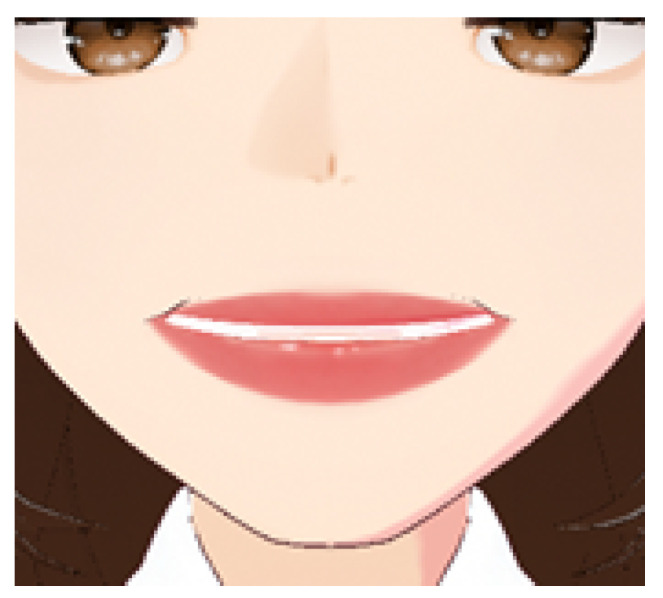
Cluster 3: Includes the letters ‘CH’, ‘E2’, ‘G2’, ‘H’, ‘NG’, ‘R’, ‘SH2’, ‘Y2’, ‘ZH’.

**Figure 7 sensors-25-03703-f007:**
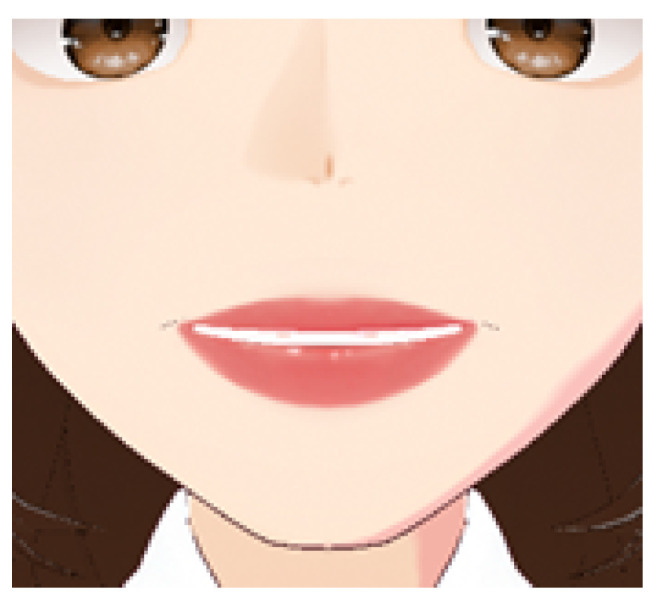
Cluster 4: Includes the letters ‘C’, ‘D’, ‘E’, ‘G’, ‘I’, ‘K’, ‘L’, ‘N’, ‘Q’, ‘S’, ‘SH’, ‘T’, ‘X’, ‘Y’, ‘Z’.

**Figure 8 sensors-25-03703-f008:**
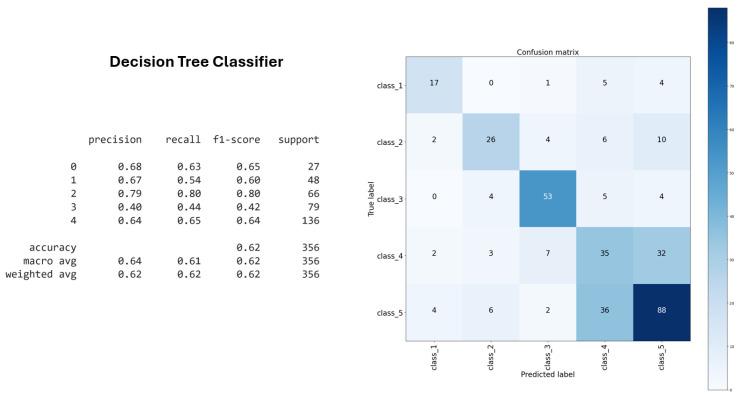
Decision Tree classifier accuracy.

**Figure 9 sensors-25-03703-f009:**
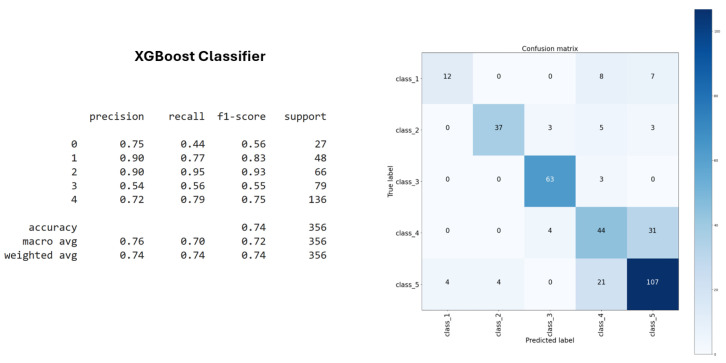
XGBoost classifier accuracy.

**Figure 10 sensors-25-03703-f010:**
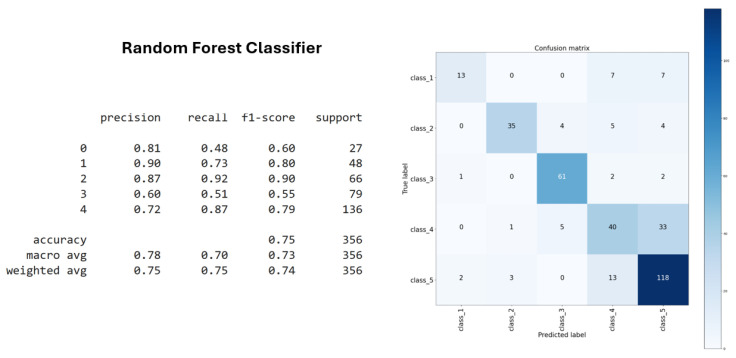
Random Forest classifier accuracy.

**Table 1 sensors-25-03703-t001:** Manual classification of fingerspelling alphabet.

Cl	Visual Characteristics of Lip Movements	Lip Movements
1	Labial sounds (lip closure): Lips touch each other.	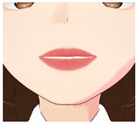
2	Dental-alveolar sounds (teeth and tongue visible): Teeth visible and tongue close to teeth.	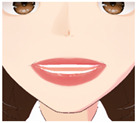
3	Central oral sounds (tongue in the middle of the mouth): Tongue in the middle of the mouth, often one tooth visible.	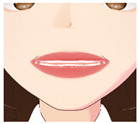
4	Open vowels (wide mouth): Mouth opens wide.	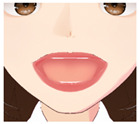
5	Extended vowels (mouth extends in both directions): Mouth opens wide and extends both ways.	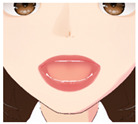
6	Smiling sounds (smile shape): Mouth takes a smile shape.	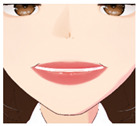
7	Rounded vowels (lip rounding with mouth opening): Lips rounded.	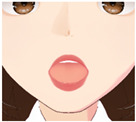
8	Rounded vowels (lip rounding with less mouth opening): Lips rounded.	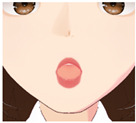
9	Raised vowels (upper lip raised): Upper lip slightly raised.	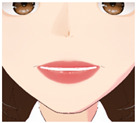
10	Fricative sounds (upper lip raised): Upper lip raised.	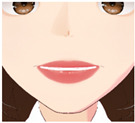

**Table 2 sensors-25-03703-t002:** Silhouette-based evaluation of clustering configurations.

Clusters	Silhouette Sc.	Best Cl.	Problematic Cl.	Comments
10	0.38	4, 5, 7, 9	0, 2	Several isolated groups; overall weak separation.
9	0.39	5, 6, 7	1, 9	Merging observed between some clusters.
8	0.40	4, 5, 6	1, 2	Compact, but partial overlap remains.
7	0.41	5, 6	0, 2	Higher internal density.
6	0.42	4, 5	0	Four clearly separated clusters.
5	0.43	3, 4	0	Simpler structure; minor blending.
4	0.44	2, 3	0	Best cohesion–separation tradeoff.

**Table 3 sensors-25-03703-t003:** Clustering outcomes based on varying cluster counts.

Number of Clusters	Observations
10	Some groups, including Cluster 1, show significant overlap; data are not well separated.
9	Clusters 5, 6, and 7 are well segmented; however, Cluster 1 remains poorly defined.
8	Improved segmentation of Clusters 4, 5, and 6; overlap persists between Clusters 1 and 2.
7	Clusters 5 and 6 display high internal density; Cluster 0 remains indistinct.
6	Clusters 3, 4, and 5 are well separated; Cluster 0 continues to lack clear boundaries.
5	Clustering quality declines; group boundaries begin to merge.
4	Achieves a better balance between boundary clarity and data homogeneity; identified as the optimal configuration.
5 (after additional splitting)	Subdivision of Cluster 1 into two sub-clusters improves classification performance.

Note. Results are based on unsupervised clustering experiments using various algorithms. Cluster separability and quality were assessed visually and through classification performance.

**Table 4 sensors-25-03703-t004:** Classification accuracy of machine learning algorithms based on clustered data.

Classification Algorithm	Accuracy
Decision Tree Classifier	62%
XGBoost Classifier	74%
Random Forest Classifier	75%

Note. The classification results confirm the validity and separability of the clusters formed through unsupervised learning. The performance of the classifiers supports the effectiveness of the clustering process and its suitability for subsequent multimodal training.

## Data Availability

The original contributions presented in this study are included in the article. The dataset and code are available at https://github.com/NurzadaEnu/Lip-reading, accessed on 8 June 2025. Further inquiries can be directed to the corresponding author.

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
