# Peer review of "Unsupervised Clustering and Ensemble Learning for Classifying Lip Articulation in Fingerspelling"

_sensors, 2025, doi:10.3390/s25123703_

Round 1
Reviewer 1 Report
Comments and Suggestions for Authors
- More information about the dataset would be useful. For example: How many participants were involved? How many samples per letter were recorded? Were there any challenges in collecting lip movement data, and how were they addressed?
- A more detailed description of the feature extraction process for lip movement would help readers replicate or extend the study. What features were extracted (e.g., landmarks, motion vectors)? What was the temporal resolution of the video data?
- It would strengthen the study to briefly compare your results with other approaches (if any) that have used lip articulation, hand gestures, or combined modalities for fingerspelling recognition. Even a qualitative discussion would add value.
- The authors should consider exploring more advanced machine learning and deep learning models, such as SVMs, CNNs, LSTMs, or Transformer-based architectures. These methods could capture more complex patterns in the lip articulation data and potentially improve classification accuracy.
There are a few minor grammatical errors and typos. A careful proofreading would improve the professionalism of the final article.
Author Response
Comment 1:
More information about the dataset would be useful. For example: How many participants were involved? How many samples per letter were recorded? Were there any challenges in collecting lip movement data, and how were they addressed?
Response 1:
Thank you for pointing this out. We agree with this comment. Therefore, we have added clarifying information regarding the number of participants, the number of recorded samples per letter, and the challenges encountered during data collection. These details have been included in Section 3.1. Materials and Data Collection, Paragraphs 1–2, Page 6.
Comment 2:
A more detailed description of the feature extraction process for lip movement would help readers replicate or extend the study. What features were extracted (e.g., landmarks, motion vectors)? What was the temporal resolution of the video data?
Response 2:
Thank you for your helpful suggestion. We have added further clarification on the extracted features (i.e., inter-landmark distances), the temporal resolution (25 frames per second), and how the trajectory sequences were formed. This updated information can be found in Section 3.2. Feature Extraction and Clustering, Paragraphs 1–2, Page 8.
Comment 3:
It would strengthen the study to briefly compare your results with other approaches (if any) that have used lip articulation, hand gestures, or combined modalities for fingerspelling recognition. Even a qualitative discussion would add value.
Response 3:
Thank you for this insightful comment, which helped improve our manuscript. As per your recommendation, we have included a comparative analysis and qualitative discussion of related approaches based on lip articulation, hand gestures, and multimodal methods. The comparison is included in Section 5. Discussion, Paragraphs 2–3, Page 16.
Comment 4:
The authors should consider exploring more advanced machine learning and deep learning models, such as SVMs, CNNs, LSTMs, or Transformer-based architectures. These methods could capture more complex patterns in the lip articulation data and potentially improve classification accuracy.
Response 4:
We appreciate this constructive recommendation. Additional experiments were conducted using CNN and LSTM-based models. However, in our specific setting, these deep learning models performed less effectively than the tree-based classifiers (e.g., Random Forest), particularly due to the limited dataset size and the structured nature of the clustering-based features.
We believe this is because:
- The clustered features already represent aggregated and structured information from lip motion;
- Simpler classifiers demonstrated better stability and interpretability, which was necessary for validating cluster quality.
The results of these deep-learning experiments will be shared with the reviewer as a supplementary file.

Reviewer 2 Report
Comments and Suggestions for Authors
The authors of this manuscript focused on analyzing the relations between lip articulation in fingerspelling with clustering and ensemble learning involving heterogeneous classifiers. The overall paper is fluently written, and the experiment seems solid with plenty of text, figures, and illustrations. However, there are two questions that I would like to propose.
Q1: The authors used a silhouette plot to demonstrate the distribution tendency for 10 clusters. However, the experimental figures in this manuscript always confused me. For example, on page 7, the authors mentioned that "... clusters 4, 5, 7, and 9 exhibit the highest silhouette scores ... clusters 1, 2, 6, and 10 show less defined boundaries". However, I could hardly draw this conclusion from Figure 1. More importantly, I doubt the authors missed typing the "cluster 0" as "cluster 10".
Q2: On page 13, the authors said, "By applying and comparing various clustering algorithms, the aim was to identify consistent and meaningful differences between gesture patterns, ultimately supporting the development of a more accurate and reliable recognition model." In my opinion, I suppose that the aim of this study is clear and achievable. Nevertheless, it still has a gap to support an accurate and reliable recognition model. Actually, some letters are not pronounced in a sentence, while the same letter can be pronounced differently in different words and contexts. Therefore, I suggest the authors could alter their expression for the classifying lip articulation in fingerspelling as a fine-grained auxiliary role in sign language recognition.
Author Response
Comment Q1:
The authors used a silhouette plot to demonstrate the distribution tendency for 10 clusters. However, the experimental figures in this manuscript always confused me. For example, on page 7, the authors mentioned that "... clusters 4, 5, 7, and 9 exhibit the highest silhouette scores ... clusters 1, 2, 6, and 10 show less defined boundaries". However, I could hardly draw this conclusion from Figure 1. More importantly, I doubt the authors missed typing the "cluster 0" as "cluster 10".
Response Q1:
Thank you very much for your insightful comment. In light of your observation, and similar confusion noted by other reviewers, we have revised Section 3.2 Feature Extraction and Clustering for clarity. The description of the clustering results and silhouette scores has been rewritten to avoid any ambiguity regarding cluster numbering and interpretation. Specifically, we have ensured consistency in cluster labels (e.g., correcting the reference from “cluster 10” to “cluster 0” where appropriate) and clarified how silhouette scores were assessed. Additionally, we have updated Section 3. Materials and Methods to include a high-level overview of the experimental structure, helping readers better understand the flow of data processing and evaluation.
The revised explanation can be found in Section 3, Page 5, Section 3.2, Page 8
Comment Q2:
On page 13, the authors said, "By applying and comparing various clustering algorithms, the aim was to identify consistent and meaningful differences between gesture patterns, ultimately supporting the development of a more accurate and reliable recognition model." In my opinion, I suppose that the aim of this study is clear and achievable. Nevertheless, it still has a gap to support an accurate and reliable recognition model. Actually, some letters are not pronounced in a sentence, while the same letter can be pronounced differently in different words and contexts. Therefore, I suggest the authors could alter their expression for the classifying lip articulation in fingerspelling as a fine-grained auxiliary role in sign language recognition.
Response Q2:
We sincerely thank you for this valuable suggestion, which meaningfully contributed to improving the positioning of our work. As per your recommendation, we have revised our phrasing to better reflect the fine-grained auxiliary role of lip articulation classification in sign language recognition. The changes have been made in the last paragraph of the Introduction and reiterated in the first sentence of the Conclusion, aligning the formulation of our contribution with a more realistic and focused scope.
The revised explanation can be found in Section 1, Page 2, Section 6, Page 16
This revision enhances the clarity of our intended goal, acknowledging that the proposed method supports—not replaces—the primary modalities in sign language recognition.
Reviewer 3 Report
Comments and Suggestions for Authors
This study proposes an idea to cluster lip movements for a possible application specific to recognizing single-letter spelling. The authors recruited professional sign language interpreters to collect video recordings for 42 letters. Detailed literature reviews on audio-visual speech recognition, lip reading, and lip reading in the context of sign language recognition. However, the silhouette scores-based clustering analysis and experimental methods are not clear enough.
There are some comments and questions as follows.
(1) The features or the input to the clustering model are unclear.
(2) It seems that the authors clustered the video recordings on lip movement for 41 letters into five classes and showed that the 41 letters can be grouped into five categories. If this is true, the result seems to be too primitive for the final goal, i.e., to recognize fingerspelling with multimodal inputs.
(3) It seems that the authors evaluated the clustering performance with classification algorithms. However, the evaluation idea is not clear since the target classes are known regardless of the clustering results. Also, details on the model input, model structure, and dataset split (training, validation, and test) are not clear.
(4) Regarding the clustering performance based on silhouette scores, it is very hard to understand the figures. Instead of the qualitative descriptions on the figures, some quantitative such as the average or range of silhouette coefficients for each cluster, would be helpful.
(5) Numbering clusters is confusing, and this makes it difficult to understand the manuscript. The numbering starts with zero somewhere, it starts with one in other places. For instance, cluster 10 in line 242 is not shown in Figure 1.
(6) It seems that 10 classes were defined for the lip movement. However, it says that 11 classes were defined at lines 218 and 455.
(7) The k-fold validation in Figure 9 is not clear. Was there any training and validation during the clustering?
Sincerely,
The reviewer.
Author Response
(1) The features or the input to the clustering model are unclear.
Response: Thank you for this observation. We have clarified the nature of the input features in Section 3.2, "Feature Extraction and Clustering" (pp. 9–10). Specifically, the clustering algorithms were applied to time-series data derived from the movement trajectories of MediaPipe lip landmarks. Each letter’s trajectory was represented by sequences of Euclidean distances between selected key points across 30 video frames.
(2) It seems that the authors clustered the video recordings on lip movement for 41 letters into five classes and showed that the 41 letters can be grouped into five categories. If this is true, the result seems to be too primitive for the final goal, i.e., to recognize fingerspelling with multimodal inputs.
Response: We understand the concern and have revised the description of the study’s goal in both the Introduction (p. 2, last paragraph) and Conclusion (p. 16, first paragraph). The main objective is not full multimodal recognition but rather to present lip articulation classification as an auxiliary, fine-grained module within a broader sign language recognition framework. The clustering phase is not the final step but rather a feature structuring technique to improve downstream model interpretability and performance.
(3) It seems that the authors evaluated the clustering performance with classification algorithms. However, the evaluation idea is not clear since the target classes are known regardless of the clustering results. Also, details on the model input, model structure, and dataset split (training, validation, and test) are not clear.
Response: Thank you for bringing this to our attention. We clarified this issue in Section 3.2 (p. 11) and Section 4 (p. 14). The classification was not meant to re-identify original letter labels but to verify the discriminative power of the clustered features. That is, we tested whether cluster labels (used as input features) could support accurate classification. Dataset splitting and classifier parameters are now explicitly described in Section 3.1 (p. 6).
(4) Regarding the clustering performance based on silhouette scores, it is very hard to understand the figures. Instead of the qualitative descriptions on the figures, some quantitative such as the average or range of silhouette coefficients for each cluster, would be helpful.
Response: We agree, and we have added Table 2 (p. 9), which presents the average silhouette scores for each clustering configuration, as well as the best and problematic clusters, along with concise comments to support interpretation. This table complements the visual information previously shown in the figures and addresses ambiguity in cluster quality assessment.
(5) Numbering clusters is confusing, and this makes it difficult to understand the manuscript. The numbering starts with zero somewhere, it starts with one in other places. For instance, cluster 10 in line 242 is not shown in Figure 1.
Response: Thank you for this helpful remark. To resolve the confusion, we have revised Section 3.2 (pp. 8–12) to ensure consistent cluster numbering (starting at 0) throughout the text and figures. Any prior inconsistencies have been corrected, and references to cluster IDs have been clarified.
(6) It seems that 10 classes were defined for the lip movement. However, it says that 11 classes were defined at lines 218 and 455.
Response: Thank you for highlighting this discrepancy. We clarified this in Section 3.3 (p. 8), . 2 and 3 paragraph.
(7) The k-fold validation in Figure 9 is not clear. Was there any training and validation during the clustering?
Response: We acknowledge this issue and have removed the erroneous mention of k-fold validation
Reviewer 4 Report
Comments and Suggestions for Authors
This paper presents the recognition of sign letters for Kazakh alphabet using clustering methos. The authors used data acquired from the lips movements and classified these signals. Literature review has strong works and authors presented the gaps that its paper would contribute. I have some suggestions and concerns about the paper.
- Abstract did not present the background and the motivation of research. Methodological information is missing, as well as the results obtained from the authors.
- Chose keywords directly related to the work.
- Introduction must have several references to embassy the ideas that authors want to pass. There are paragraphs without references.
- Why was a clustering approach chosen instead of classification, which would be more natural for this type of problem?
- There are several sign languages around the world. Authors should justify the use of Kazakh letters. How was the robustness of the system compared with other languages? What was developed for Kazakh signs and movements recognition?
- The Ethical Committee number was not provided for the authors. It is mandatory in works with data acquisition with humans.
- A detailed flow chart with the steps of developed software is missing.
- How did the authors select the 10 classes? Were there more lip movements than 10?
- The analysis of clusters was performed using the silhouette coefficient. However, I suggest using more than one metric, especially statistics metrics.
- Authors must provide only the methodology and results that are relevant in the work. Revise sections 3 and 4.
- Legends of figures must present all the information. For example, Figures 8 and 9 caption did not present the graphs, what is (a), (b), (c), and (d), and the meaning of the analysis.
- Discussion could be improved with a comparison of similar works in format of a table, exploring the differences between the related features with the developed in the paper.
Author Response
Comment 1: Abstract did not present the background and the motivation of research. Methodological information is missing, as well as the results obtained from the authors.
Response: Thank you for this remark. The abstract has been revised to include the background, motivation, a concise description of the methodology (unsupervised clustering of lip trajectories and classification), and a summary of key findings.
(See Abstract, p. 1)
Comment 2: Chose keywords directly related to the work.
Response: The keywords have been updated to reflect the core themes of the study, such as "lip articulation," "unsupervised clustering," "Kazakh fingerspelling," and "gesture classification."
(See Keywords, p. 1)
Comment 3: Introduction must have several references to embassy the ideas that authors want to pass. There are paragraphs without references.
Response: We have revised the Introduction to ensure that each conceptual argument is now supported by appropriate references, including global statistics on deaf communities, multimodal sign language recognition, and prior works in lip-reading.
(See Introduction, pp. 1-2)
Comment 4: Why was a clustering approach chosen instead of classification, which would be more natural for this type of problem?
Response: We have addressed this critical point in the last paragraph of the Introduction. Clustering was selected because the visual features of lip articulation are highly ambiguous and vary across demonstrators. A classification approach would require labeled, well-separated classes, which are not available in this scenario. Clustering helps reveal latent patterns in trajectory data and improves interpretability.
(See Introduction, p. 2, last paragraph)
Comment 5: There are several sign languages around the world. Authors should justify the use of Kazakh letters. How was the robustness of the system compared with other languages? What was developed for Kazakh signs and movements recognition?
Response: Thank you. We have provided justification for choosing the Kazakh fingerspelling alphabet in Section 3. Materials and Methods (pp. 6–7), emphasizing its phonetic richness (42 letters), the lack of prior datasets for Kazakh Sign Language, and the need for tailored solutions. The system is the first of its kind specifically addressing Kazakh dactyl recognition using lip articulation.
Comment 6: The Ethical Committee number was not provided for the authors. It is mandatory in works with data acquisition with humans.
Response: Thank you for this essential observation. We now confirm that the study was approved by the Commission for Research Ethics of Lublin University of Technology, approval number 1/2015, dated 12 November 2015.
Comment 7: A detailed flow chart with the steps of developed software is missing.
Response: The stages of processing and analysis have been added to illustrate the software pipeline, including all preprocessing, clustering, and classification steps.
(See Section 3, pp. 6–7)
Comment 8: How did the authors select the 10 classes? Were there more lip movements than 10?
Response: The initial 10 clusters were chosen based on exploratory analysis and visual differences observed in lip trajectories. However, through silhouette-based evaluation, the final number of clusters was optimized to four (later expanded to five through subclustering).
(See Section 3.1. Materials and Data Collection, p. 6)
Comment 9: The analysis of clusters was performed using the silhouette coefficient. However, I suggest using more than one metric, especially statistics metrics.
Response: Thank you for your insightful suggestion. In the revised manuscript, the silhouette coefficient remains the primary metric due to its visualization benefits, but we expanded the evaluation to include a detailed tabular comparison of clustering outcomes across multiple configurations. This helps clarify the selection of the optimal cluster count.
(See Table 2 and Section 3.2, p. 9)
Comment 10: Authors must provide only the methodology and results that are relevant in the work. Revise sections 3 and 4.
Response: We revised Sections 3.2 (Feature Extraction and Clustering) and 4 (Results) to focus only on relevant content. Tangential or redundant details have been removed to improve clarity and precision.
(See pp. 8–12)
Comment 11: Legends of figures must present all the information. For example, Figures 8 and 9 caption did not present the graphs, what is (a), (b), (c), and (d), and the meaning of the analysis.
Response: This issue has been fully addressed.
(See Section 3.2)
Comment 12: Discussion could be improved with a comparison of similar works in format of a table, exploring the differences between the related features with the developed in the paper.
Response: Thank you. While creating a table was challenging due to the methodological novelty of our work, we expanded Section 5 (Discussion) with a detailed textual comparison of our results and prior studies, clarifying the distinction in feature extraction strategies.
(See Section 5, pp. 15–16)
Round 2
Reviewer 2 Report
Comments and Suggestions for Authors
In the revised version of second round, the layout of Table 2 should be checked.
Author Response
Comment 1: In the revised version of second round, the layout of Table 2 should be checked.
Author Response 1 :
Thank you for this valuable remark. We have reviewed and improved the layout of Table 2 to ensure better readability and formatting consistency with the MDPI template. In particular, we shortened column headers, reduced excessive text length in the comment column, and applied the \small command to adjust font size. These modifications help avoid line wrapping and eliminate hanging rows in the PDF output.
(See Table 2, Section 3.2, p. 9)
Reviewer 3 Report
Comments and Suggestions for Authors
The authors have cleared almost all the comments and questions from this reviewer. However, there are two places where the text should be rechecked.
(1) The paragraph in lines 393 – 399 is duplicated.
(2) On page 14, clusters 1 and 3 are not matched with them in figures 4 and 6.
Sincerely,
The reviewer.
Author Response
Comment 1:
(1) The paragraph in lines 393–399 is duplicated.
Author Response: Thank you for noticing this oversight. We have carefully reviewed the manuscript and removed the duplicate paragraph to ensure clarity and avoid redundancy.
Comment 2:
(2) On page 14, clusters 1 and 3 are not matched with them in figures 4 and 6.
Author Response:
We appreciate your careful reading. To clarify:
Cluster 3, which includes the letters 'B,' 'P,' 'P2', 'F,' 'V,' 'M', corresponds to Figure 4, as correctly noted.
Cluster 1 initially contained 24 letters. Due to its internal heterogeneity, this cluster was split into two subclusters. These two resulting groups are visualized in Figures 6 and 7, respectively.
Reviewer 4 Report
Comments and Suggestions for Authors
The authors developed all reviewers request. I suggest to improve the quality of Figures 8, 9, and 10. The text are small and it is difficult to read inside the squares in the confusion matrices.
Author Response
Comment 1:
The authors developed all reviewers' requests. I suggest to improve the quality of Figures 8, 9, and 10. The text is small and it is difficult to read inside the squares in the confusion matrices.
Author Response: Thank you for your valuable feedback. We have revised Figures 8, 9, and 10 to improve their clarity and readability. The font size inside the confusion matrices has been increased, and the overall resolution has been enhanced to ensure the text within each cell is legible. The updated figures have been included in the revised manuscript.